


# Assessment of Flood Susceptibility Using Support Vector Machine in the Belt and Road Region

Jun Liu[1], Junnan Xiong[1,2*], Weiming Cheng[2,3], Yi Li[4], Yifan Cao[1], Yufeng He[1], Yu Duan[1], Wen He[1], Gang Yang[1]

[1]School of Civil Engineering and Geomatics, Southwest Petroleum University, Chengdu, 640500, China

[2]State Key Laboratory of Resources and Environmental Information System, Institute of Geographic Sciences and Natural Resources Research, CAS, Beijing,100101, China

[3]University of Chinese Academy of Sciences, Beijing 100049, China

[4]Aerospace Information Research Institute, Chinese Academy of Sciences, Beijing, 100094, China

*Correspondence to*: Junnan Xiong (xiongjn@swpu.edu.cn); Jun Liu (201922000655@stu.swpu.edu.cn); Weiming Cheng (chengwm@lreis.ac.cn); Yi Li (liyi@aircas.ac.cn); Yifan Cao (201822000588@stu.swpu.edu.cn) Yufeng He (201922000656@stu.swpu.edu.cn); Yu Duan (201922000658@stu.swpu.edu.cn); Wen He (201922000663@stu.swpu.edu.cn); Gang Yang (201922000664@stu.swpu.edu.cn).

**Abstract.** Floods have occurred frequently all over the world. During 2000-2020, nearly half (44.9%) of global floods occurred in the Belt and Road region because of its complex geology, topography, and climate. However, the degree of flood susceptibility of each sub-region and country in the Belt and Road region remains unclear. Here, based on 11 flood condition factors, the support vector machine (SVM) model was used to generate a flood susceptibility map. Then, we introduced the flood susceptibility

comprehensive index (FSCI) for the first time to quantify the flood susceptibility levels of the sub-regions and countries in the Belt and Road region. The results reveal the following. (1) The SVM model used in this study has an excellent accuracy, and the AUC values of the success-rate curve and prediction-rate curve were higher than 0.9 (0.917 and 0.934 respectively). (2) The areas with the highest and high flood susceptibility account for 12.22% and 9.57% of the total study area respectively, and these areas are

mainly located in the southeastern part of Eastern Asia, almost the entirely of Southeast Asia and South Asia. (3) Of the seven sub-regions in the Belt and Road region, Southeast Asia is most susceptible to flooding and has the highest FSCI (4.49), followed by South Asia. (4) Of the 66 countries in this region, 16 of the countries have the highest flood susceptibility level (normalized FSCI > 0.8) and 5 countries





(normalized FSCI > 0.6) have a high flood susceptibility level. These countries need to pay more attention

to flood mitigation and management. The above findings provide useful information for decision-making

in flood management in the Belt and Road region. In the future study, higher quality flood points, and

climate change factors should be considered.

**Keywords:** Food susceptibility; Machine learning; Support vector machine (SVM); The Belt and Road

region

**1 Introduction**

Various natural disasters occur frequently worldwide, among which flooding is the most common and

devastating (Stefanidis and Stathis, 2013). Both society and ecosystems suffer from the profound effects

of floods. This is reflected in the loss of lives and property and the changes in the natural environment,

respectively (Hirabayashi et al., 2013). According to recent estimates, the economic losses caused by

floods around the word account for up to 40% of the total losses caused by all natural disasters (Xia et

al., 2008). In the Belt and Road region, 1483 floods occurred from 2000 to 2020, accounting for 44.9%

of the total floods around the world based on the statistic from the Emergency Disasters Database (EM-

DAT, CRED, http://www.emdat.be/). For example, the New Asia-Europe Continental Bridge, the

Bangladesh-China-India-Myanmar Economic Corridor, the China-China South Economic Corridor and

the China-Pakistan Economic Corridor have all been threatened by flooding for a long time (Lei et al.,

2018). Unfortunately, most of the countries in the Belt and Road region are developing countries with

underdeveloped economies and weak disaster resilience, and thus, they lack material reserves and

emergency relief capabilities for disasters response (Cui et al., 2018). Thus, the countries in this region

always suffer more severe losses in the face of disasters. According to the EM-DAT, disaster losses in

the countries in the Belt and Road region are more than twice the global average. The mortality rate of

disaster victims in this region is also much higher than the global average, and in South and Southeast

Asian countries, it is even 10 times higher than the global average (Ge et al., 2020). However, the

construction of "the Belt and the Road" involves a large number of infrastructure and major engineering

projects in transportation, communication and energy. These projects are always planned and deployed

in disaster-prone areas, and these areas are highly concentrated in less developed countries. Frequent

flooding poses a major threat to offshore investments, project safety and regional development in these



areas (Ge et al., 2020). Even worse, due to global climate change, both the frequency and intensity of the

floods in this area are expected to increase in the future (Temmerman et al., 2013). Within this context,

it is extremely important to establish a scientific basis for flood prevention and disaster reduction in the

Belt and Road region.

Although it is still considered impossible to prevent flooding completely, an accurate flood susceptibility

map would enable us to predict the locations where floods may occur. An assessment based on this map

could effectively help to relieve the impacts and losses caused by floods (Ali et al., 2019) by allowing

people to respond to flooding in an anticipatory rather than a reactive manner (Zhao et al., 2018). As a

semi-quantitative method, flood susceptibility assessment considers the comprehensive influences of the

disaster-inducing factors and hazard-inducing environment, and it has been widely applied in flood

insurance, floodplain management and disaster warning systems (Hallegatte et al., 2013;Zou et al., 2013).

There are four main types of methods for developing flood susceptibility maps, including multi-criteria

decision analysis (MCDA) methods, statistical methods, physically based models, and machine learning

methods. Because of simplicity of MCDA methods, they have been widely used in flood susceptibility

assessment, e.g., the analytic hierarchy process (AHP) (Lyu et al., 2020;Santos et al., 2019;Tang et al.,

2018). However, this type of method relies heavily on the judgment of experts, which makes the results

somewhat subjective and uncertain (Chowdary et al., 2013). Statistical methods, generally include

bivariate statistical analysis (BSA) and multivariate statistical analysis (MSA) (Tehrany et al., 2014).

Among the BSA methods, the frequency ratio (FR) is one of the most common methods used to quantify

the impact of each class of factors on flooding (Jebur et al., 2014). In contrast, logistic regression (LR)

determines the influence of each individual flooding factor as a typical MSA method (Jebur et al., 2014).

These statistical analysis methods have been confirmed to have an excellent performance in flood

susceptibility assessment (Rahmati et al., 2015). However, they rely on predicted variables that are based

on linear assumptions, while flooding generally has a non-linear structure (Tehrany et al., 2015b).

Physically based models are efficient for flood inundation modeling (Dimitriadis et al., 2016). For

examples, one-dimensional model such as the Mike 11 and two-dimensional models such as the SRH-

2D are frequently used (Knebl et al., 2005;Lavoie and Mahdi, 2017). These physically based models

have the ability to descript the details of flooding including the flood inundation extent, water depth, and

the velocity (Mazzoleni et al., 2013). However, the drawback of these physically based models is obvious,

which is that requiring a sea of input data and substantial computational resources (Tehrany et al., 2019).


For machine learning methods, due to their improvement in recent years, their use in flood susceptibility assessment has become increasingly common. Random forest (RF) (Wang et al., 2015), artificial neural networks (ANN) (Li et al., 2013), support vector machines (SVM) (Tehrany et al., 2015b), and decision

tree (DT) (Tehrany et al., 2013) are popular machine learning algorithms. These machine learning methods can solve non-linear problems better, but their accuracy is extremely dependent on the quality of the sample points. Distinctly, each of the above four methods has inherent advantages and disadvantages. Thus, at present, there is no consensus on which type of model should be applied to a given scenario and which model is best (Khosravi et al., 2018a). Taking into account the characteristics

of the study area and the availability of data, in this study, the SVM model was used to generate a flood susceptibility map. The excellent generalization of the SVM (Bahram et al., 2019) was one reason for selecting this method. In addition, as a machine learning method, the SVM not only avoids the subjective determination of weights as occurs in MCDA methods, but it is also does not require a large number of model parameters compared with physically based models.

For the analysis of the flood susceptibility results, most studies (Wang et al., 2015;Zhang and Chen, 2019;Hu et al., 2017b) only semi-quantitatively assessed the proportion and distribution of the flood susceptibility classes. These assessments cannot provide a fully quantitative representation of the degree of flood susceptibility in a region, so more in-depth studies are needed to quantify the flood susceptibility level of each region. To this end, in this study, the flood susceptibility comprehensive index (FSCI) is

introduced to quantify the flood susceptibility level of each country and sub-region in the study area based on the concept and calculation method of the ecological vulnerability synthesis index (EVSI) (Tian, 2018).

In this study, we divided the Belt and Road region into 627,454 0.1°×0.1° grids and used each grid as a research unit to assess the flood susceptibility. Then, a flood susceptibility map of the study area was

generated using the SVM model. Based on this, the main purposes of the current study are as follows: (1) analyzing the spatial pattern of the areas prone to flooding in the Belt and Road region; and (2) evaluating the flood susceptibility levels of countries and sub-regions in the Belt and Road region by calculating the FSCI.





## 2 Materials

### 2.1 Study area

To strengthen the ties between Asia, Europe, and Africa three continents and form a human community with a shared density, "The Belt and Road" Initiative was proposed by China in 2015 (Jiang et al., 2018). The Belt and Road region (Fig. 1) spans Asia, Africa, and Europe, encompassing three continent and 66 countries (including Kashmir). It contains a population of about 4.4 billion people and has a combined gross domestic product (GDP) of 2.3 billion dollars, accounting for 63% and 29% of the world totals, respectively (Zhang, 2018). The study area is highly undulatory, with altitudes ranging from −438 m to 8,728 m. In addition, landforms are also complex, including mountains, hills, valleys, plateaus and several other types of terrains (Yu et al., 2019). There are eight types of climate in this region, including both monsoon and continental climate characteristics (Zhou et al., 2020).The precipitation, is spatially heterogeneous. The annual mean total precipitation in this region during 2000 – 2018 increased from 0.92 mm in the southwest to 6067.71 mm in the southeast. In conclusion, due to the vast area and complex geology, topography and climate in the region, favorable disaster-conditions have been formed. These conditions lead to the occurrence of diversity, frequency and severe natural disasters in this region. Among which, flooding is the most frequent. According to the statistic provides by the EM-DAT, of the 3483 natural disasters that occurred in the Belt and Road region from 2000 to 2020, 1438 were floods, accounting for about 41.3% of all disasters. Therefore, it is of great significance to conduct flood susceptibility assessment in this region.

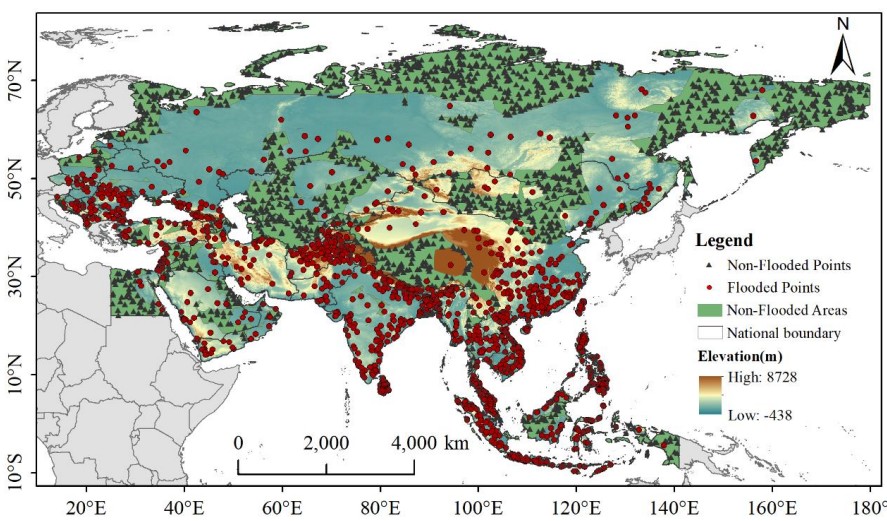




**Figure 1: The distribution of flood sample points in the study area.**

**2.2 Data**

**2.2.1 Flood inventory map**

To apply the machine learning method to predicting the areas where floods may occur in the future, existing flood records are needed as a training reference (Khosravi et al., 2018b). For the global scale, two flood datasets are commonly used, the International Disaster Database (EM-DAT) and the Global

Active Archive of Large Flood Events. These two datasets provide various information about floods so that people can better understand the impacts of floods (Sampson et al., 2015). The flood location dataset used in this study was obtained from the Global Active Archive of Large Flood Events, Dartmouth Flood Observatory, University of Colorado (http://floodobservatory.colorado.edu/), which has been supported by the National Aeronautics and Space Administration (NASA) and used in several studies around the

world (Li et al., 2019).

For the Belt and Road region, this study selected 1,500 flooded points from January 2000 to March 2020 as the sample dataset. Based on the information in this sample dataset, first we extracted the areas where no flood has occurred (Fig. 1). Then, the same number of non-flooded points as the number of flooded points were randomly selected in the non-flooded areas, and values of 1 and 0 were assigned to the

flooded and non-flooded points, respectively. Finally, all of the sample points were randomly divided, with 70% used as training points and 30% used as verification points.

**2.2.2 Flood conditioning factors**

In flood susceptibility mapping, the first task is to construct a spatial database that contains the flood condition factors. However, the suitable flood condition factors vary with the characteristics of the

different areas (Tehrany et al., 2013), and the same factors have very different influences in different areas (Kia et al., 2012). After comprehensive consideration of the actual characteristics in the study area, the review of relevant studies (Mahmoud and Gan, 2018;Ali et al., 2020), and the availability of data, a total of 11 factors that are closely related to flood disasters were chosen for use in this study. The selected factors include the maximum three-day precipitation (M3DP), altitude (AL), standard deviation of

elevation (SDE), slope (SL), flow accumulation (FA), topographic wetness index (TWI), river density (RD), fractional vegetation cover (FVC), percentage of impervious surface (PIS), land cover (LC), and soil texture (ST). Table 1 presents the primary sources for the factors layers used in this study. Each flood





condition factor was converted into a grid database with a spatial resolution of 0.1×0.1° in ArcGIS 10.6
and was normalized using the maximum normalization method. These normalized flood condition factors

are shown in Fig. 2.

**Table 1: Primary sources for the factor layers used in this study.**

| Classification | Sub-Classification | Source of Data | Time | Resolution |
|---|---|---|---|---|
| Flood inventory map | Flood inventory map | Dartmouth Flood Observatory (http://floodobservatory.colorado.edu/) | 2000−2020 | / |
| DEM | Altitude Standard deviation of elevation Slope Flow accumulation Topographic wetness index | SRTM (http://srtm.csi.cgiar.org/srtmdata/) | 2010 | 1×1 km |
| GPM | Maximum three-day precipitation | Goddard Earth Sciences Data and Information Services Center (https://pmm.nasa.gov/precipitation-measurement-missions) | 2000−2018 | 0.1°×0.1° |
| River | River density | Open-Street-Map (https://www.openstreetmap.org/) | 2019 | 1:50,000 |
| MCD12Q1 | Land cover | LAADS DAAC (https://ladsweb.modaps.eosdis.nasa.gov/search/) | 2015 | 0.5×0.5 km |
| MOD13Q1 | Fractional vegetation cover | LAADS DAAC (https://ladsweb.modaps.eosdis.nasa.gov/search/) | 2015 | 0.5×0.5 km |
| Impervious surface | Percentage of impervious surface | GHSL (https://ghslsys.jrc.ec.europa.eu/) | 2014 | 30×30 m |
| Soil texture | Soil texture | FAO (http://www.fao.org/) | 2008 | 1×1 km |



(a)   Maximum three-day precipitation(M3DP)

In particular, heavy rainfall with a short duration has a great potential for flooding (Ali et al., 2020).

Many studies (Liu et al., 2017;Huang and Zhang, 2016) have shown that M3DP has a non-negligible influence on the occurrence of floods. The possibility of flooding is considered to increase with increasing M3DP. In the current study, the M3DP factor map (Fig. 2a) was calculated using Global Precipitation Measurement (GPM) data, which records the average daily precipitation everywhere in the world from 2000 to 2018.

(b)   Altitude (AL)

Altitude is also an important factor affecting the occurrence of flood disasters. It is usually inversely related to flood susceptibility since water flows from higher elevations to lower elevations (Mohamoud, 1992;Vojtek and Vojtekova, 2019). In general, the higher the altitude, the less prone the area is to flooding. In this study, the altitude (Fig. 2b) was represented by the digital elevation model (DEM), which was

derived from Shuttle Radar Topography Mission (SRTM) data.

(c)   Standard deviation of elevation (SDE)

The standard deviation of elevation reflects the degree of topographic variation within a certain range (Zhou et al., 2000). The undulations in the topography directly affect the gathering of the water flow, thus affecting the occurrence of floods. Generally, the susceptibility to flooding decreases as the degree

of topographic undulation increase (Zhou et al., 2000). In this study, the SDE (Fig. 2c) was obtained by calculating the elevations of 25 grids (including itself) in the 5 × 5 neighborhood around a grid.

(d)   Slope (SL)

Slope is given a higher priority in flood sensitivity mapping (Zaharia et al., 2017). The size of the slope has a significant influence on the surface runoff, soil erosion, and vertical percolation (Samanta et al.,

2018). Therefore, the slope can affect the occurrence of flooding. Generally, floods occur more frequently in low-slope areas. In contrast, high-slope areas have fast water flow, resulting in a low permeability and high runoff (Chapi et al., 2017), so floods in these areas are relatively rare. In this study, the slope (Fig. 2d) was calculated using the DEM and the slope calculation tool in ArcGIS.

(e)   Flow accumulation (FA)

Flow accumulation may be one of the most important factors in assessing flood susceptibility (Kazakis et al., 2015). Flow accumulation refers to the sum of the water flow from surrounding units and paths,





which leads to increased flow in a specific unit, and it helps identify the convergence area of the surface runoff (Mahmoud and Gan, 2018). Thus, the flow accumulation has a positive effect on the occurrence of floods. In this study, the FA (Fig. 2e) was obtained through a series of calculations in ArcGIS.

(f) Topographic wetness index (TWI)

Several previous research studies have reported that the TWI is a meaningful factor for the study of flood susceptibility (Ali et al., 2020). Its role is to quantify the topographical control over hydrological processes. In other words, the TWI measures the impact of the topography on runoff generation(Das, 2018). According to previous studies (Regmi et al., 2010), the TWI (Fig. 2f) was calculated using the

following equation:

$$TWI = \ln(A_s / \tan \beta) \tag{1}$$

where $A_s$ is the specific catchment area ($m^2 m^{-1}$) and $\beta$ (radian) is the slope gradient (in degrees).

(g) River density (RD)

River density is also one of the direct factors affecting flooding (Ali et al., 2020). It is defined as the ratio

of the length of the river network to the area in a unit. Generally, the higher the drainage density of an area, the greater the likelihood of flooding (Tehrany et al., 2015a). After downloading river network data covering the entire study area from Open-Street-Map (https://www.openstreetmap.org/), we calculated the river density (Fig. 2g) in ArcGIS.

(h) Fractional vegetation cover (FVC)

For floods, the fractional vegetation cover is usually considered to be one of the most important influencing factors (Khosravi et al., 2016). It expresses the status of the vegetation coverage. The larger the FVC value, the higher the degree of vegetation coverage. Compared with areas with high FVCs, areas with low FVCs are more prone to flooding (Tehrany et al., 2013). The FVC values (Fig. 2h) used in this study were calculated from normalized difference vegetation index (NDVI) data, which were

downloaded from the Level-1 and Atmosphere Archive and Distribution System Distributed Active Archive Center (LAADS DAAC). The following equation was used for the calculations (Zhang et al., 2017):

$$FVC = (NDVI - NDVI_{soil}) / (NDVI_{veg} - NDVI_{soil}) \tag{2}$$

where $NDVI_{soil}$ is the bare land NDVI and $NDVI_{veg}$ is the vegetation NDVI value of a full vegetation




coverage area. According to the previous experience, we selected the statistical 90% NDVI value as $NDVI_{veg}$ and the 5% NDVI value as the $NDVI_{soil}$.

(i)   Percentage of impervious surfaces (PIS)

The percentage of impervious surfaces has been used in flood risk assessment studies(Hu et al., 2017a) because it has a certain impact on the occurrence of floods. The impervious surfaces affect the vertical

percolation of water flow. In general, the larger the percentage of impervious surfaces, the more prone the area is to water accumulation, leading to flooding. The impervious surface data used in this study were obtained from the Global Human Settlement (GHSL, https://ghslsys.jrc.ec.europa.eu/), and the percentage of impervious surfaces (Fig. 2i) in each grid was calculated using the Zonal Statistics tool in ArcGIS.

(j)   Land cover (LC)

The type of land cover (Fig. 2j) changes the surface runoff to a certain extent, thereby affecting the occurrence of floods (Bui et al., 2019). The land cover data used in this study were obtained from LAADS DAAC (https://ladsweb.modaps.eosdis.nasa.gov/search/) and are for 2015. In order to quantify the impacts of the various types of land cover on floods, we used the information value method to

calculate their contributions to flooding. The results (Table 2) were calculated using Eq. (3).

(k)   Soil texture (ST)

Soil texture (Fig. 2k) has a relatively obvious impact on the occurrence of floods (Peng et al., 2019). The texture is a property of soil that describes the relative proportion of the different grain sizes in the soil (Wang et al., 2015). The soil texture data used in this study were downloaded from the Food and

Agriculture Organization of the United Nations (FAO, http://www.fao.org/). We used the same approach as that described above to quantify the impact of soil texture on flooding, and the results are shown in Table 2.

**Table 2: Information values of various types of land cover and soil texture.**

| Factor | Category | Ni | Si | I |
|---|---|---|---|---|
| | Evergreen Needleleaf Forest | 2 | 11592 | -2.27 |
| | Evergreen Broadleaf Forest | 153 | 21553 | 1.44 |
| | Deciduous Needleleaf Forest | 1 | 2974 | -1.61 |
| Land cover | Deciduous Broadleaf Forest | 27 | 11931 | 0.30 |
| | Mixed Forests | 20 | 46981 | -1.37 |
| | Closed Shrublands | 0 | 463 | 0.00 |



|  | | | |
|---|---|---|---|
| Open Shrublands | 18 | 79584 | -2.00 |
| Woody Savannas | 170 | 66092 | 0.43 |
| Savannas | 93 | 60489 | -0.09 |
| Grasslands | 132 | 131631 | -0.51 |
| Permanent Wetlands | 4 | 6720 | -1.03 |
| Croplands | 250 | 76921 | 0.66 |
| Urban Areas | 35 | 1872 | 2.41 |
| Cropland -Natural Vegetation Mosaic | 64 | 4434 | 2.15 |
| Snow and Ice | 2 | 3602 | -1.10 |
| Barren or Sparsely Vegetated | 73 | 93570 | -0.76 |
| Water Bodies | 6 | 6845 | -0.65 |
| Clay(heavy) | 5 | 196 | 2.72 |
| Silty clay | 3 | 4687 | -0.96 |
| Clay | 146 | 45773 | 0.64 |
| Silty clay loam | 1 | 833 | -0.33 |
| Clay loam | 18 | 31638 | -1.08 |
| Soil texture    Silt loam | 34 | 83006 | -1.41 |
| Loam | 525 | 253981 | 0.21 |
| Sandy clay loam | 189 | 38810 | 1.07 |
| Sandy loam | 79 | 81744 | -0.55 |
| Loam sand | 37 | 67381 | -1.11 |
| Sand | 13 | 19205 | -0.91 |

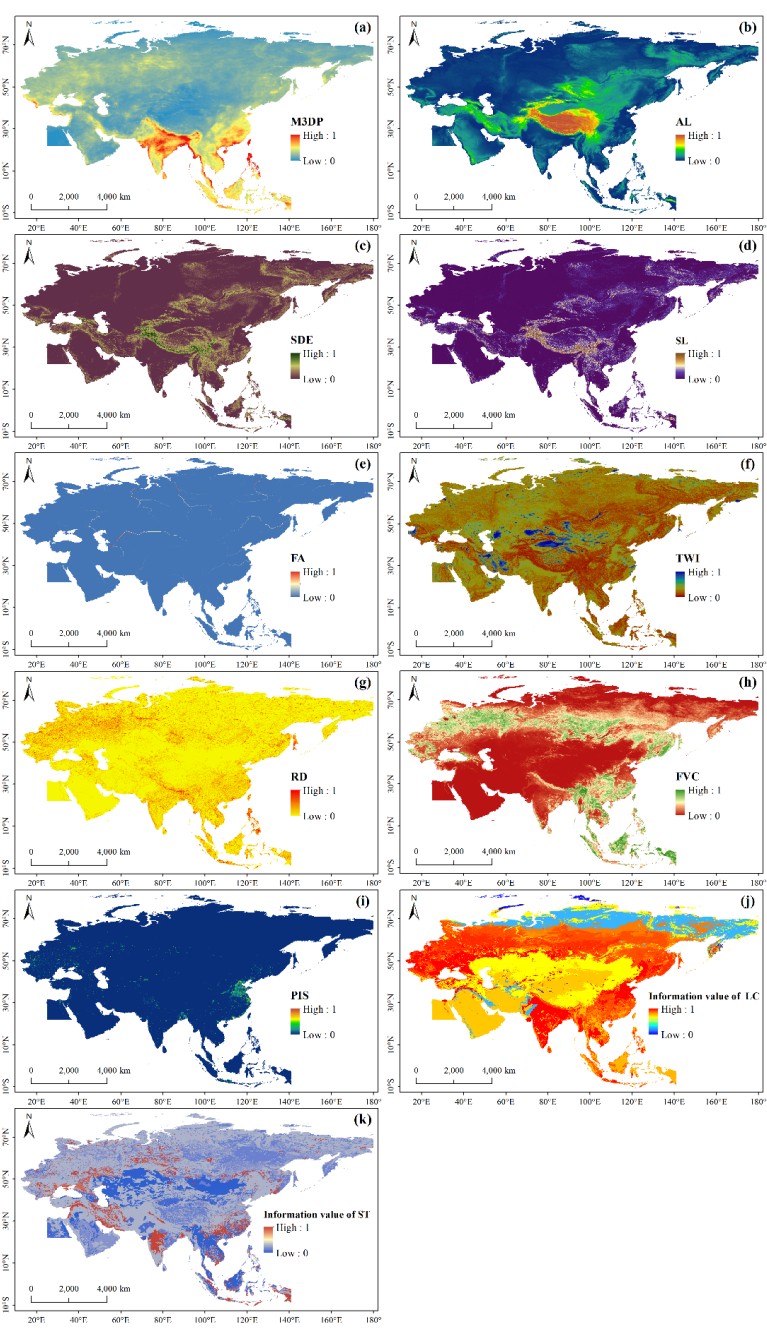


**Figure 2: Normalized flood condition factors: (a) Maximum three-day precipitation (M3DP), (b) Altitude (AL), (c) Standard deviation of elevation (SDE), (d) Slope (SL), (e) Flow accumulation (FA), (f) Topographic wetness index (TWI), (g) River density (RD), (h) Fractional vegetation cover (FVC), (i) Percentage of impervious surface (PIS), (j) Information value of Land cover (LC), (k)**
**Information value of Soil texture (ST).**



## 3 Methods

### 3.1 Data processing methods

#### 3.1.1 Information value method

The information value method is an indirect statistical method (Du et al., 2017), which is frequently used

in landslide sensitivity assessment, but it is relatively new in flood sensitivity mapping. The purpose of

the information method is to determine the weight of each factor (Sarkar et al., 2013). Inspired by this, it

was used to determine the weight of each category of land cover and soil texture in this study. The method

was originally proposed by Yin and Yan (1988) and was slightly modified by Van Westen (1993) (Sarkar

et al., 2013), which was shown as follow:

$$I = \ln \frac{N_i / S_i}{N / S} \qquad (3)$$

where $I$ is the weight of factor class $i$; $N_i$ is the number of floods in class $i$; $S_i$ is the number of pixel class

$i$; $N$ is the number of floods in the whole study area; $S$ is the number of pixels in the entire study area.

#### 3.1.2 Correlation analysis of conditioning factors

If there is a high correlation between variables, the model estimation will be distorted or difficult to

estimate accurately (Zhang XD et al., 2018). Usually, several methods such as the Pearson correlation

coefficient method, the variance decomposition ratio, the conditional index, and variance inflation factor

(VIF) and tolerance are used to quantify the correlations between factors (Khosravi et al., 2018b). In this

study, we used the VIF and tolerance to measure the relationships between the 11 factors. When the VIF

is greater than 10 or the tolerance is less than 0.1, the factor has multiple collinearity problems and should

be eliminated. Otherwise, there is no collinearity between the factors.

### 3.2 Support Vector Machine (SVM)

The SVM is one of the most popular machine learning algorithms. It is a supervised learning binary

classifier based on the structural risk minimization principle (Yao et al., 2008). Because of its nonlinear

mathematical structure, the complex nonlinear relationship between the inputs and outputs in a system

can be represented by the SVM (Li et al., 2016). Generally, there are two methods of constructing an

SVM model. The first is to construct an optimum linear separating hyperplane, which is used to separate


the data patterns. The second is to use the kernel function to convert the original nonlinear data pattern

into a linearly separable format in the high-dimensional feature space (Yao et al., 2008). The major steps

of the algorithm are as follows:

(1)    Assume that $T = \{x_1, x_2, ..., x_n, y\}$ is the training set of known samples where $x_i$ is the $i_{th}$ input data,

and $y$ is the output data where $i = 1, 2, ..., n$.

(2)    Separate the training set into two categories using an n-dimensional hyperplane to obtain the

maximum interval:

$$\frac{1}{2}\|w\|^2 \tag{4}$$

$$\text{Subject to} \quad y_i\left((w \cdot x_i) + b\right) \geq 1 \tag{5}$$

where ‖$w$‖ is the norm of the hyperplane normal; $b$ is a scalar base, and (·) represents the product

operation.

(3)    Using the Lagrange multiplier, the cost function can be defined as follows:

$$L = \frac{1}{2}\|w\|^2 - \sum_{i=1}^{n} \lambda_i \left( y_i((w \cdot x_i) + b) - 1 \right) \tag{6}$$

where $\lambda_i$ is the Lagrangian multiplier. By using standard procedures, the solution can be obtained by

minimizing the duality of $w$ and $b$ using Equation (6) (Vapnik, 1995).

(4)    For the non-separable case, the constraints can be modified by introducing slack variables, $\xi_i$

(Vapnik, 1995):

$$y_i\left((w \cdot x) + b\right) \geq 1 - \xi_i \tag{7}$$

Thus Equation (6) becomes:

$$L = \frac{1}{2}\|w\|^2 - \frac{1}{vn}\sum_{i=1}^{n} \xi_i \tag{8}$$

where $v \in (0,1]$, which is introduced in order to account for misclassifications (Xu et al., 2012).

The selection of the kernel type retains the significance for performance and the results of the SVM

(Damaševičius, 2010). At present, the linear kernel (LN), polynomial kernel (PL), radial basis function

(RBF) kernel, and sigmoid kernel (SIG) are the most commonly used kernel types. Several studies have

shown that the BRF has a better performance in geological disaster prediction, which is the reason of

selecting it in this study (Kia et al., 2012;Pradhan, 2012). The BRF is described as follow:





$$K\left(x_i, x_j\right) = e^{-\gamma(x_i - x_j)^2}, \gamma > 0 \tag{9}$$

where $\gamma$ is the parameter of the kernel function. Sometimes kernel functions are parameterized using $\gamma = 1/2\sigma^2$, where $\sigma$ is an adjustable parameter that governs the performance of the kernel.

### 3.3 Model validation method

The receiver operating characteristics (ROC) curve has been used to evaluate the performances of models in many studies (Wang et al., 2015;Tehrany et al., 2014;Huang et al., 2020). For each possible critical

value, the ROC is considered to be a graphical representation of the trade-off between the false negative (X-axis) rate and the false positive (Y-axis) rate (Pourghasemi and Beheshtirad, 2015). It is executed by using the area under ROC (AUC) to compare the known data on flooding with acquired flooding probability map. The value of the AUC is between 0 (a diagnostic test that cannot distinguish between floods and non-floods) and 1 (Bahram et al., 2019). Generally, the greater the AUC, the higher the

accuracy of the model. The relationship between the performance of a model and the AUC can be classified into the following categories: 0.9–1 (excellent), 0.8–0.9 (very good), 0.7–0.8 (good), 0.6–0.7 (moderate), and 0.5–0.6 (poor). In this study, 70% of the chosen flood locations were used to obtain the success-rate curve and 30% of the chosen flood locations were used to obtain the prediction-rate curve, which can reflect the goodness of fit and prediction power of the SVM model, respectively (Termeh et

al., 2018).

### 3.4 Flood Susceptibility Comprehensive Index (FSCI)

In order to reflect the degree of flood susceptibility in the study area more intuitively and comprehensively, in this study the flood susceptibility comprehensive index (FSCI) was calculated for each region and country by referring to the calculation method for the ecological vulnerability synthesis

index (EVSI) (Tian, 2018). The calculation method is as follows:

$$CIFS = \sum_{i=1}^{n} p_i \times \frac{A_i}{S} \tag{10}$$

where $FSCI$ is the flood susceptibility comprehensive index of a country; $P_i$ is the class value of the $i_{th}$ flood susceptibility calss; $A_i$ is the areas of $i_{th}$ flood susceptibility class; and $S$ is the total area of the country. In this study, the lowest, low, moderate, high, and highest flood susceptibility classes correspond

to $P_i$ values of 1, 2, 3, 4, and 5, respectively.



### 3.5 Workflow of flood susceptibility assessment

The workflow of the flood susceptibility assessment is illustrated in Fig. 3. First, a set of available data, containing a flood inventory map and flood condition factors, was collected from different sources. For the condition factors, the information value method was used to quantify the weights of discrete factors

(LC and ST), and the maximum value normalization method was used to normalize the information values of the two discrete factors and the original value of continuous factors. Then, variance inflation factors (VIF) and tolerances were used to verify that there was no serious collinear relationship between the indicators. The results of the factor correlation test are shown in Table 3. As can be seen from Table 3, the SDE has the lowest tolerance (0.142) and the highest VIF (7.065). However, neither of them exceed

the critical values (0.1 and 10, respectively) indicating that there is no serious collinearity among the 11 factors. Therefore, all 11 factors were input into the SVM model for the training and prediction steps to obtain the flood susceptibility map. After classifying the map into five classes: lowest, low, moderate, high, and highest using the equal interval method, we calculated the FSCI of each country. Based on the FSCI, the flood susceptibilities of countries were classified into five levels also using the equal interval

method. Finally, we validated the accuracy of the SVM model and analyzed the results in terms of the spatial patterns and flood susceptibility level of each country and region. It should be noted that the SVM model was conducted using the e1071 package in R software.

**Table 3: Results of the collinearity statistics.**

| Flood condition factors | Collinearity statistics | |
|---|---|---|
| | Tolerance | VIF |
| Percentage of impervious surface (PIS) | 0.833 | 1.200 |
| Altitude (AL) | 0.575 | 1.739 |
| Flow accumulation (FA) | 0.982 | 1.018 |
| Land cover (LC) | 0.558 | 1.792 |
| River density (RD) | 0.704 | 1.421 |
| Maximum three-day precipitation (M3DP) | 0.463 | 2.159 |
| Slope (SL) | 0.163 | 6.123 |
| Soil texture (ST) | 0.829 | 1.206 |
| Standard deviation of elevation (SDE) | 0.142 | 7.065 |
| Topographic wetness index (TWI) | 0.929 | 1.077 |
| Fractional Vegetation Cover (FVC) | 0.574 | 1.743 |


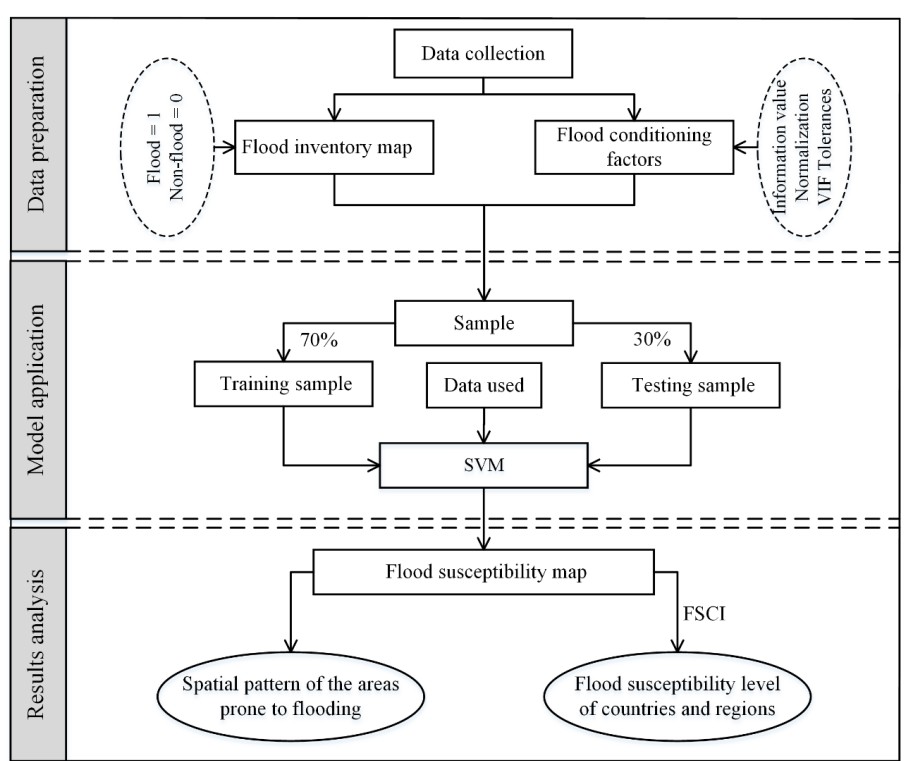


**Figure 3: Workflow of flood susceptibility mapping conducted in this study.**

**4 Results and discussion**

**4.1 Accuracy assessment**

The success-rate curve and the prediction-rate curve of the SVM model are shown in Fig. 4a and 4b,

respectively. According to Fig. 4a, the AUC of the success-rate curve of the SVM model is 0.917, which

indicates that the model has an excellent goodness of fit. For the prediction-rate curve of SVM (Fig. 4b),

the AUC is 0.934, indicating that the SVM model has a good prediction effectiveness. Overall, both the

AUC values of the success-rate curve and the prediction-rate curve of SVM were greater than 0.9, which

demonstrates that the results obtained in this study using the SVM model are scientific and reliable.


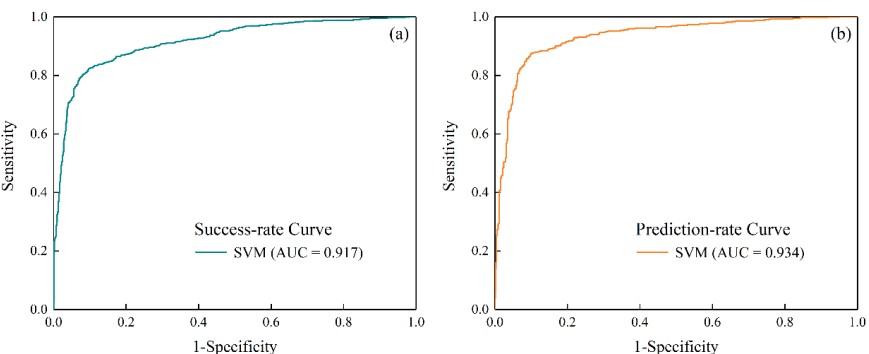

Figure 4: Validation of SVM model: (a) Success-rate Curve, (b) Prediction-rate Curve.

**4.2 Classification results of flood susceptibility map**

Figure 5 shows the flood susceptibility map obtained using the SVM model in this study, and Table 4 presents the area percentages of the various susceptibility levels in the Belt and Road region. According to the statistics (Table 4), the lowest flood susceptibility zone accounts for 32.91% of the study area. The low, moderate, and high flood susceptibility zones account for 31.56%, 13.74% and 9.57% of the study area respectively; and highest flood susceptibility area accounts for 12.22%. Although more than half of the study area is in lowest and low flood susceptibility zones, accounting for about 64.47% together, nearly 1/5 of the study area has the high or highest flood susceptibility, with an area of approximately $1,103.70 \times 10^4 \, \mathrm{km}^2$, which is the focus of our attention.

In terms of spatial distribution pattern of the susceptibility (Fig. 5), the areas with high and highest flood susceptibility are mostly distributed in the southeastern part of Eastern Asia, almost the entirety of Southeast Asia and of South Asia. Thus, Asia is the part of the study area suffering the most from the floods, which is consistent with the results of Kundzewicz et al. (Kundzewicz et al., 2014). In addition, several coastal areas in Europe, located in the Mediterranean climate zone, also have high or highest flood susceptibilities. The northwestern part of Eastern Asia, the entirety of Central Asia, and Northern Asia mainly have low and lowest flood susceptibilities.

However, this spatial distribution pattern is somewhat difference from the results of flood risk assessment of global watersheds conducted by Li et al. (Li et al., 2019). The differences are mainly reflected in the classifications of the flood susceptibilities in Europe and Northern Asia. The European region mainly has a moderate flood susceptibility in our study, while it has the highest class in the results of Li et al; Northern Asia region mainly has lowest flood susceptibility in this study, while it has the highest class in

the results of Li et al. According to Li et al., the reason for these differences is that the methods of

selecting the non-flooded points are not the same. In this study, we fully excavated the information in the

flood inventory map to identify the areas that have not experienced flooding (Fig. 1) and selected non-

flooded points in these areas. However, Li et al. selected non-flooded points in the deserts and ice fields,

which may make the conditions for non-flooding more severe. When comparing with the locations of the

flood points in Fig. 1, we found that the flood susceptibilities determined in their study may have been

overestimated, which was also mentioned by the authors. In conclusion, the results of this study have a

certain degree of improvement compared to the results of previous studies.

**Table 4: Area percentages (%) of the various susceptibility levels in the Belt and Road region.**

| Flood susceptibility class | Area($10^4 \times km^2$) | Coverage(%) |
|---|---|---|
| Lowest | 1666.41 | 32.91 |
| Low | 1598.18 | 31.56 |
| Moderate | 695.73 | 13.74 |
| High | 484.56 | 9.57 |
| Highest | 619.16 | 12.22 |

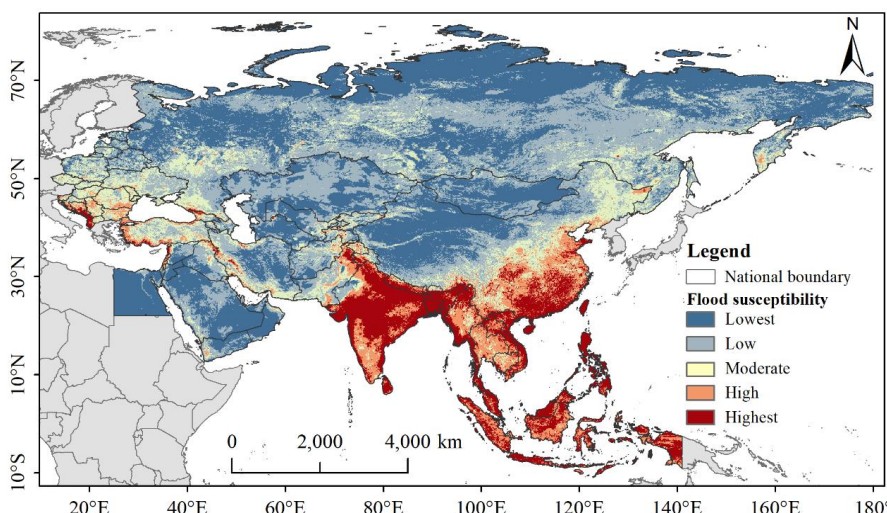

**Figure 5: Flood susceptibility map obtained using the SVM model.**

**4.3 FSCIs of the different regions and countries**

**4.3.1 The entire Belt and Road Region**

In this study, the entire region was divided into seven sub-regions, including Eastern Asia, Southeast Asia,

South Asia, Central Asia, Western Asia (including Egypt), Central-Eastern Europe (CEE) and Russia (Fig.





6), and their respective FSCI values were calculated (Table 5). As can be seen from Table 5, Southeast

Asia has the highest FSCI (4.49), followed by South Asia with the FSCI (4.17), both of which are much

greater than the overall FSCI value (2.37) of the study area. This result illustrates that Southeast Asia is

the most flood-prone region in the Belt and Road region, followed by South Asia. Apparently, Western

Asia, Central Asia, and Russia are the least likely to experience flooding, with FSCI values of 1.83, 1.67,

and 1.62, respectively.

The results of the FSCI values of each country are presented in Table 6 and Fig. 7. As can be seen, 16

countries have the highest flood susceptibility, including Brunei, Maldives, Bangladesh, Philippines,

Albania, Sri Lanka, Malaysia, Vietnam, Montenegro, Laos, Timor-Leste, India, Indonesia, Singapore,

Nepal, and Myanmar. The proportions of the areas with the highest flood susceptibility exceed 50% in

all of these countries. In addition, 5 countries have the high flood susceptibilities; 16 countries have

moderate flood susceptibilities; 17 countries have low flood susceptibilities; and 12 countries have the

lowest flood susceptibility. Clearly, of the 66 countries in the study area, 21 countries have high or highest

flood susceptibility levels, accounting for 31.8%.

**Table 5: FSCI values of the regions in the Belt and Road region.**

| Region Name | Proportion of the area of each flood susceptibility class (%) | | | | | FSCI |
| --- | --- | --- | --- | --- | --- | --- |
| | Lowest | Low | Moderate | High | Highest | |
| Southeast Asia | 0.34 | 0.55 | 5.27 | 37.17 | 56.67 | 4.49 |
| South Asia | 2.13 | 10.19 | 10.87 | 22.22 | 54.59 | 4.17 |
| CEE | 9.79 | 32.20 | 45.46 | 10.23 | 2.32 | 2.63 |
| Eastern Asia | 35.36 | 26.55 | 17.22 | 12.13 | 8.74 | 2.32 |
| Western Asia | 45.89 | 32.52 | 15.18 | 5.19 | 1.22 | 1.83 |
| Central Asia | 41.25 | 51.98 | 5.79 | 0.88 | 0.10 | 1.67 |
| Russia | 48.52 | 41.34 | 9.72 | 0.39 | 0.03 | 1.62 |
| The Belt and Road Region | 32.91 | 31.56 | 13.74 | 9.57 | 12.22 | 2.37 |

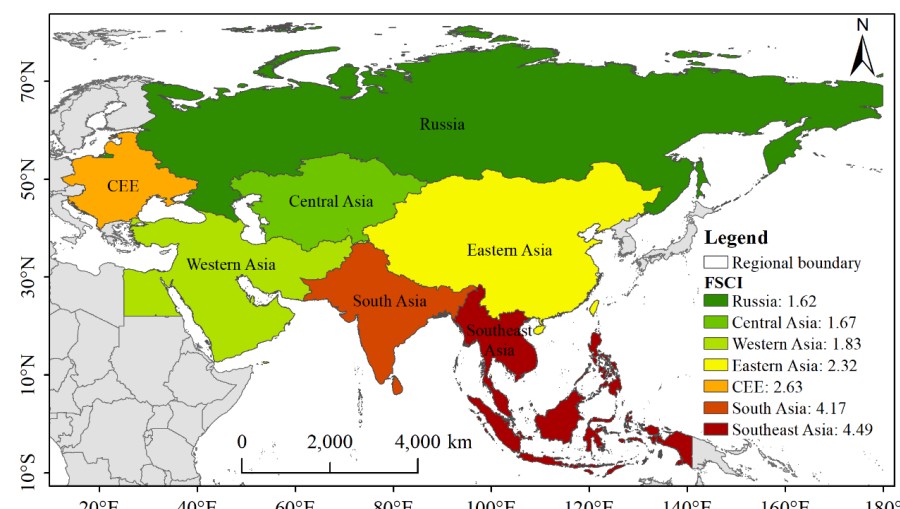


**Figure 6: FSCIs of the seven sub-regions in the Belt and Road region.**

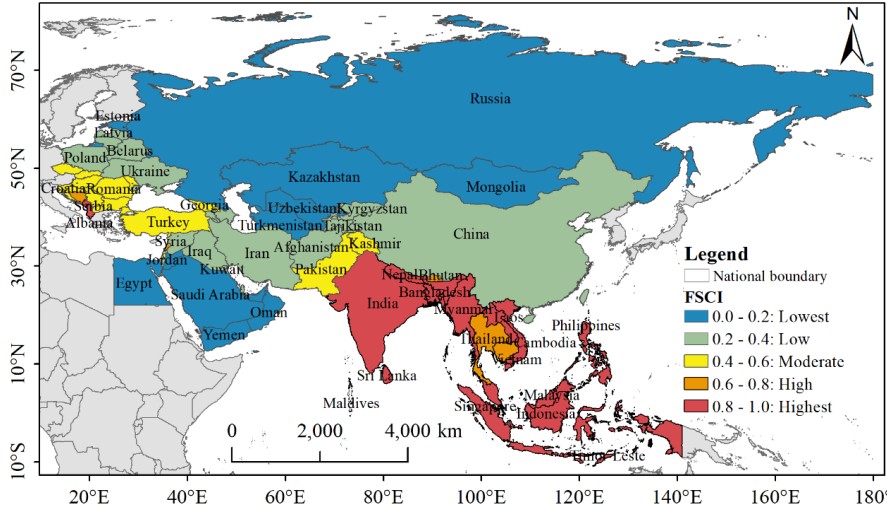

**Figure 7: Levels of FSCI of the countries in the Belt and Road region.**

### 4.3.2 Russia

Due to the vast size of Russia, it was analyzed as a separate region in this study. It has been pointed out

that the number of floods has increased in both the Asian part of Russia (Northern Asia) and the European

part (Frolova et al., 2017), so a flood susceptibility assessment for Russia is of great value. According to

the results of the FSCI calculations (Table 5), Russia is the region least threatened by flooding in the

study area, with the lowest FSCI value of 1.62. The lowest and low flood susceptibility zones occupy

48.63% and 41.25% of the total area of Russia, respectively. Still, almost 10% of the country has a





moderate flood susceptibility, mainly in the southern part of its European part and in the central and southeastern parts of its Asian part, which is in good agreement with the results of Frolova et al. (Frolova et al., 2017). In terms of the flood condition factors, the results of this assessment were mainly dependent on the distribution of the M3DP. In short, Russia is not a country with a high susceptibility to flooding,

but certain areas still face a certain threat from flooding. It should be noted that several studies (Frolova et al., 2017;Shalikovskiy and Kurganovich, 2017) have shown that the main cause of flooding in Russia is snowmelt, followed by rainfall. However, due to the limited research conditions, snowmelt was not considered in this study, and more attention should be paid to this issue in subsequent studies.

### 4.3.3 Eastern Asia

Table 5 shows that Eastern Asia has a relatively low FSCI value (2.32), with of the low and lowest flood susceptibility zones accounting for 35.36% and 26.55% of the area, respectively. Still, the high and highest flood susceptibility zones account for 20.87% of the area. More interestingly, as is shown in Fig. 5, there is a clear regional variation in flood susceptibility in Eastern Asia, decreasing from southeast to northwest. The southeastern part of Eastern Asia mainly has high or highest flood susceptibilities, while

the northwestern part has low or lowest flood susceptibilities. This pattern was also reported by Liu et al. (Liu et al., 2017). The causes of this phenomenon can be analyzed from two perspectives (the factors and the climate). From the perspective of the factors, the factors (M3DP, RD, LC, and ST) that drive flooding in the southeastern part of Eastern Asia have high values, while the values of these factors are low in the northwestern part of the region. In addition, the factors that are negatively correlated with flood

probability (e.g., SDE and SL) have low values in the southeast and high values in the northwest. From the perspective of climate, the Southeastern of Eastern Asia is located in the subtropical monsoon climate zone, which is influenced by the Eastern Asian summer winds (Ding et al., 2020). Therefore, it is prone to extreme rainstorms, which in turn cause floods. The northwestern part of Eastern Asia is located in the temperate continental climate zone, with is dry and experiences little precipitation, so flooding does not

easily occur. These above two aspects contribute to the decreasing susceptibility of flooding in Eastern Asia from southeast to northwest. The two countries in Southeast Asia (China and Mongolia), Table 6 and Fig. 7, have low and lowest flood susceptibility level, respectively. However, because of the vast size of China and its high flood-proneness in the southeast, China still needs to devote more energy to dealing with floods. In conclusion, Eastern Asia faces a certain degree of flood threat, and its flood-prone areas



are concentrated in Southeastern China, while Northwestern China and Mongolia are less prone to flooding.

### 4.3.4 Southeast Asia

Southeast Asia is a key maritime transportation route in the Belt and Road Strategy, as well as a major concentrator to the economies in Asia. Table 5 shows that Southeast Asia has the highest FSCI value

(4.67) of the seven sub-regions, with the high and highest flood susceptibility zones accounting for 37.17% and 56.67% of its area, respectively. Thus, it can be concluded that Southeast Asia is the most flood-prone region in the Belt and Road region, which is consistent with the results of An et al. (An et al., 2020). By analyzing the spatial distribution of the flood susceptibility (Fig. 5), we found that almost the entirety of Southeast Asian region has a high or highest flood susceptibility. This spatial distribution can be

explained in two ways. Geographically, since the western Pacific region is the main origin of typhoons, the probability of flooding due to typhoon rainstorms is significantly higher in Southeast Asia facing the Pacific than in other regions (Zhou, 1995). In terms of flood condition factors, the M3DP and RD values in Southeast Asia are high, while SL and AL values are low, together making Southeast Asia prone to flooding. According to the FSCIs of each country (Table 6 and Fig. 7), among the 11 countries in

Southeast Asia, 9 countries including Brunei, Philippines, Malaysia, Vietnam, Laos, Timor-Leste, Indonesia, Singapore and Myanmar, have the highest flood susceptibility level, and 2 countries including Thailand and Cambodia, have the high flood susceptibility level. Clearly, all of the countries in Southeast Asia are highly susceptible to flooding and require more investment in and effort towards flood mitigation is required.

### 4.3.5 South Asia

South Asia is an essential component of the Belt and Road region since it connects China with Western Asia, Africa, and even Europe. Table 5 shows that South Asia has a high FSCI value of 4.17, which is second only to Southeast Asia. The high flood susceptibility and highest flood susceptibility zones account for 22.22% and 54.59% of the area, respectively. Therefore, South Asia is considered to be the

second most flood-prone area in the Belt and Road region. As be seen from Table 6 and Fig. 7, among the eight countries and territories in South Asia (including Kashmir), except for Kashmir and Pakistan, which have moderate flood susceptibilities, and Bhutan, which has a high flood susceptibility, the other 5 countries including Maldives, Bangladesh, Sri Lanka, India, and Nepal, are have the highest flood


susceptibility level. Thus, the countries in South Asian also face a great possibility of flooding, which
       has also been reported by Abbas et al. (Abbas et al., 2016). Explanations for the flood-proneness of South
       Asia can be analyzed from two perspectives. In terms of climate, common causes (precipitation,
       snowmelt, etc.), and continental factors (the El Niño-Southern Oscillation) are responsible for the
       susceptibility South Asia to flooding. However, among these causes, precipitation is the most important
       factor for flooding in the countries of South Asia because they are heavily influenced by monsoon

weather systems (Mirza, 2011). In terms of flood indicators, the M3DP also echoes this view, with the
       M3DP in South Asia indicating a high level of risk. In addition, the higher LC, ST, and RD values and
       the lower AL, SL, FVC, and SDE values in South Asia jointly contribute high flood susceptibility of
       South Asia. Even worse, studies have shown that the frequency, magnitude and extent of floods in South
       Asia may increase due to climate warming (Mirza, 2011). Therefore, flood prevention and mitigation in

the countries in South Asia will be a long-term and arduous task.

       **4.3.6 Western Asia**

       Geographically, Western Asia is located at the junction of three continents, Asia, Europe, and Africa, and
       it is also the key hub of the Atlantic and Indian oceans, connecting the three continents (Han and Zhou,
       2014). As can be seen from Table 5, the FSCI values of Western Asia is low (1.83), with the low or lowest

flood susceptibility zones accounting for 93.23% of it area. Therefore, Western Asia is perceived to be
       less threatened by flooding. However, the high or highest flood susceptibility zones still account for 6.41%
       of the area, which are mainly distributed in coastal areas (including the Mediterranean coast and the
       Persian Gulf coast). This can be explained from two perspectives. From the viewpoint of the factors, the
       factors such as M3DP, RD, and TWI that promote flooding have low values, while the factors such as

AL, SL, and SDE that inhibit flooding have high values, which together result in the low flood
       susceptibility in Western Asia. From a climatic point of view, due to subtropical high pressure, most of
       Western Asia has a desert climate, while the northern part has a temperate continental climate. Both
       climates are characterized by low rainfall, so flooding does not easily occur in this region (Yang et al.,
       2016). However, the Mediterranean coast in the northwestern part of Western Asia, with its abundant

rainfall influenced by the Mediterranean climate, is area of high flood susceptibility. The results of the
       FSCI calculations (Table 6 and Fig. 7) show that of all the 20 countries in Western Asian part of the study
       area, 15 have low or lowest flood susceptibility levels. However, Palestine, which is on the Mediterranean





coast, has a high flood susceptibility level, and four countries, (Turkey, Georgia, Palestine and Bahrain) have a moderate flood susceptibility level. As can be seen, the vast majority of the countries in Western
Asia are less prone to flooding.

### 4.3.7 Central Asia

Central Asia is located in the core hinterland of Eurasia. It is characterized by relatively backward economic development and limited disaster preparedness (Yuan and Wang, 2015), so it is meaningful to analyze the flood susceptibility of this area. Table 5 shows that Central Asia has the second lowest FSCI
value of 1.83 (higher than only Russia), with the high and highest flood susceptibility zones accounting for less than 1% of the total area of the region, making it one of the least flood-prone regions in the Belt and Road region. As is shown by the spatial distribution of the flood susceptibility (Fig. 5), almost all of Central Asia has low or lowest flood susceptibility classes. The reasons for these results can be explained from two perspectives. In terms of the factors, although AL, SL, and SDE values are low, the M3DP
value is also low. So, there is a lack of flood-causing precipitation, thus leading to a low flood susceptibility throughout almost all of Central Asia. In terms of climate, Central Asia is a typical arid and semi-arid region, with a primarily temperate continental climate. As a result, precipitation is scarce here, (Wang, 2019) making this area less prone to flooding. As can be seen from the FSCI results (Table 6 and Fig. 7), among the five countries in Central Asia, three countries (Turkmenistan, Kazakhstan, and
Uzbekistan) have the lowest flood susceptibility level. Kyrgyzstan and Tajikistan have a low flood susceptibility level. Apparently, all five countries in Central Asia can worry less about flooding.

### 4.3.8 CEE

Despite significant investments in flood prevention, flooding remains a serious problem throughout Europe (Kundzewicz et al., 2014), so an assessment of the flood susceptibility in the European region of
the Belt and Road region is of great necessity. As is shown in Table 5, the moderate flood susceptibility zone accounts for 45.46% of the CEE, followed by the low flood susceptibility zone (32.20%), and the high or highest flood susceptibility zones still account for 12.55% of the area. Therefore, the FSCI value of CEE is also moderate (2.63), indicating that the CEE suffers from some degree of flood susceptibility. As can be seen in Fig. 5, the CEE region mainly has a moderate flood susceptibility. However, the flood
susceptibility of the CEE region has a spatial distribution pattern of decreasing from south to north. The southern Mediterranean coastal region has high or highest flood susceptibilities while the northern part




has low or lowest susceptibilities. This result is spatially consistent with the spatial distribution of the number of large floods in Europe from 1985 to 2009 made by Kundzewicz et al. (Kundzewicz et al., 2013). Based on the FSCI results (Table 6), more than half of the 19 countries in the CEE region of the

study area have a moderate level of flood susceptibility. However, two countries (Albania and Montenegro) have the highest level of flood susceptibility, and only six countries have low or lowest flood susceptibility levels. As can be seen from Fig. 7, the three countries with high or highest flood susceptibility levels are all adjacent to the Mediterranean Sea and are influenced by the Mediterranean climate. As Marchi et al. pointed out, most of the storm events in Europe were occur in the Mediterranean

and the Alpine-Mediterranean Regions (Marchi et al., 2010). Overall, the CEE countries are relatively prone to flooding, especially those near the Mediterranean coast.

**Table 6: Results of the FSCI calculations for each country.**

| ID | Country Name | Proportions of the areas with each flood susceptibility class (%) | | | | | FSCI (n) | Levels |
|---|---|---|---|---|---|---|---|---|
| | | Lowest | Low | Moderate | High | Highest | | |
| 1 | Brunei | 0.00 | 0.00 | 0.00 | 0.00 | 100.00 | 1.00 | 5 |
| 2 | Maldives | 0.00 | 0.00 | 0.00 | 0.00 | 100.00 | 1.00 | 5 |
| 3 | Bangladesh | 0.98 | 0.24 | 0.00 | 0.00 | 98.78 | 0.99 | 5 |
| 4 | Philippines | 0.90 | 1.88 | 0.41 | 7.59 | 89.22 | 0.96 | 5 |
| 5 | Albania | 0.33 | 0.00 | 2.31 | 13.53 | 83.83 | 0.95 | 5 |
| 6 | Sri Lanka | 0.37 | 0.00 | 0.19 | 17.76 | 81.68 | 0.95 | 5 |
| 7 | Malaysia | 0.22 | 0.45 | 0.64 | 19.80 | 78.89 | 0.94 | 5 |
| 8 | Vietnam | 0.22 | 0.11 | 1.62 | 24.08 | 73.97 | 0.93 | 5 |
| 9 | Montenegro | 0.67 | 0.00 | 3.33 | 20.00 | 76.00 | 0.93 | 5 |
| 10 | Laos | 0.00 | 0.00 | 1.22 | 36.92 | 61.86 | 0.90 | 5 |
| 11 | Timor-Leste | 0.00 | 1.57 | 0.00 | 39.37 | 59.06 | 0.89 | 5 |
| 12 | India | 0.93 | 2.88 | 3.94 | 24.03 | 68.22 | 0.89 | 5 |
| 13 | Indonesia | 0.41 | 0.69 | 1.84 | 39.45 | 57.61 | 0.88 | 5 |
| 14 | Singapore | 0.00 | 16.67 | 0.00 | 0.00 | 83.33 | 0.87 | 5 |
| 15 | Nepal | 0.00 | 1.34 | 11.29 | 24.22 | 63.15 | 0.87 | 5 |
| 16 | Myanmar | 0.34 | 0.45 | 11.47 | 34.16 | 53.58 | 0.85 | 5 |
| 17 | Thailand | 0.18 | 0.11 | 12.44 | 61.61 | 25.64 | 0.78 | 4 |
| 18 | Lebanon | 1.02 | 15.31 | 8.16 | 35.71 | 39.80 | 0.74 | 4 |
| 19 | Bosnia and Herzegovina | 0.00 | 1.56 | 27.08 | 47.22 | 24.13 | 0.73 | 4 |
| 20 | Cambodia | 0.07 | 0.33 | 24.42 | 59.46 | 15.73 | 0.72 | 4 |
| 21 | Bhutan | 1.16 | 12.50 | 40.99 | 16.57 | 28.78 | 0.64 | 4 |
| 22 | Serbia | 0.00 | 1.61 | 59.20 | 37.79 | 1.41 | 0.59 | 3 |
| 23 | Croatia | 1.06 | 15.61 | 38.79 | 32.42 | 12.12 | 0.59 | 3 |
| 24 | Bulgaria | 0.16 | 6.23 | 54.31 | 39.13 | 0.16 | 0.57 | 3 |

| 25 | Kashmir | 3.17 | 19.47 | 35.72 | 25.71 | 15.92 | 0.57 | 3 |
|---|---|---|---|---|---|---|---|---|
| 26 | Slovenia | 0.00 | 12.13 | 52.30 | 28.87 | 6.69 | 0.57 | 3 |
| 27 | Macedonia | 0.00 | 7.25 | 67.39 | 23.91 | 1.45 | 0.54 | 3 |
| 18 | Romania | 0.62 | 10.71 | 63.74 | 24.57 | 0.36 | 0.53 | 3 |
| 29 | Moldova | 0.00 | 6.97 | 80.35 | 12.69 | 0.00 | 0.51 | 3 |
| 30 | Turkey | 0.75 | 26.09 | 45.84 | 21.70 | 5.63 | 0.50 | 3 |
| 31 | Georgia | 9.16 | 23.30 | 32.46 | 23.43 | 11.65 | 0.50 | 3 |
| 32 | Palestine | 7.27 | 27.27 | 30.91 | 29.09 | 5.45 | 0.49 | 3 |
| 33 | Hungary | 0.36 | 16.41 | 71.89 | 11.24 | 0.09 | 0.48 | 3 |
| 34 | Bahrain | 16.67 | 33.33 | 0.00 | 50.00 | 0.00 | 0.45 | 3 |
| 35 | Slovakia | 1.01 | 20.88 | 75.25 | 2.86 | 0.00 | 0.44 | 3 |
| 36 | Pakistan | 6.88 | 37.67 | 30.35 | 18.83 | 6.26 | 0.44 | 3 |
| 37 | Czech Republic | 3.75 | 24.04 | 66.02 | 5.98 | 0.20 | 0.43 | 3 |
| 38 | China | 29.38 | 25.44 | 20.30 | 14.45 | 10.42 | 0.37 | 2 |
| 39 | Ukraine | 2.88 | 48.32 | 46.27 | 2.54 | 0.00 | 0.36 | 2 |
| 40 | Tajikistan | 10.53 | 47.03 | 31.04 | 10.12 | 1.28 | 0.35 | 2 |
| 41 | Israel | 35.81 | 20.00 | 11.16 | 31.16 | 1.86 | 0.35 | 2 |
| 42 | Kyrgyzstan | 13.15 | 43.26 | 37.44 | 5.68 | 0.47 | 0.33 | 2 |
| 43 | Poland | 17.24 | 36.89 | 43.52 | 2.32 | 0.02 | 0.32 | 2 |
| 44 | Armenia | 11.29 | 57.37 | 28.53 | 2.82 | 0.00 | 0.30 | 2 |
| 45 | Kuwait | 6.75 | 73.01 | 14.11 | 4.91 | 1.23 | 0.29 | 2 |
| 46 | Lithuania | 22.69 | 41.48 | 35.07 | 0.76 | 0.00 | 0.27 | 2 |
| 47 | Azerbaijan | 15.29 | 60.40 | 21.56 | 2.42 | 0.33 | 0.27 | 2 |
| 48 | Afghanistan | 23.82 | 50.94 | 20.79 | 4.18 | 0.27 | 0.25 | 2 |
| 49 | Iraq | 26.10 | 50.71 | 16.18 | 5.68 | 1.33 | 0.25 | 2 |
| 50 | Iran | 32.78 | 37.86 | 22.48 | 5.92 | 0.97 | 0.25 | 2 |
| 51 | Syria | 33.70 | 42.93 | 16.30 | 4.24 | 2.83 | 0.24 | 2 |
| 52 | Qatar | 12.12 | 80.81 | 5.05 | 2.02 | 0.00 | 0.23 | 2 |
| 53 | Belarus | 30.55 | 50.57 | 18.57 | 0.32 | 0.00 | 0.21 | 2 |
| 54 | Latvia | 34.15 | 44.33 | 21.00 | 0.53 | 0.00 | 0.21 | 2 |
| 55 | Estonia | 40.17 | 41.18 | 18.36 | 0.29 | 0.00 | 0.18 | 1 |
| 56 | Uzbekistan | 44.22 | 41.33 | 12.87 | 1.54 | 0.04 | 0.17 | 1 |
| 57 | Russia | 48.63 | 41.25 | 9.71 | 0.39 | 0.03 | 0.14 | 1 |
| 58 | Kazakhstan | 40.54 | 57.63 | 1.71 | 0.12 | 0.00 | 0.14 | 1 |
| 59 | Yemen | 52.64 | 37.03 | 7.89 | 2.23 | 0.21 | 0.14 | 1 |
| 60 | United Arab Emirates | 48.92 | 47.38 | 2.93 | 0.46 | 0.31 | 0.12 | 1 |
| 61 | Turkmenistan | 64.56 | 30.71 | 4.10 | 0.32 | 0.30 | 0.09 | 1 |
| 62 | Saudi Arabia | 65.60 | 32.00 | 2.24 | 0.16 | 0.00 | 0.08 | 1 |
| 63 | Oman | 72.46 | 20.94 | 5.02 | 1.42 | 0.15 | 0.07 | 1 |
| 64 | Mongolia | 66.49 | 32.31 | 1.17 | 0.02 | 0.00 | 0.07 | 1 |
| 65 | Jordan | 82.40 | 13.56 | 2.73 | 1.31 | 0.00 | 0.04 | 1 |
| 66 | Egypt | 93.91 | 5.39 | 0.65 | 0.06 | 0.00 | 0.00 | 1 |

*1 represents the lowest flood susceptibility level; 2 represents the low flood susceptibility level; 3



represents the moderate flood susceptibility level; 4 represents the high flood susceptibility level; 5 represents the highest flood susceptibility level, and FSCI(n) represents the normalized FSCI.

### 4.4 The implications and limitations

The results generated by this study not only identified the flood-prone areas in the Belt and Road region but also assessed the level of flood susceptibility of each country, and thus the results of this study provide important information for the mitigation of damages resulting from future floods. In addition, because the research on global flood susceptibility maps is relatively rare, the successful application of the model and index system used in this study provides a reference for large-scale flood susceptibility research.

Although this study has achieved reasonable results, the following limitations still exist. (1) For the methods, the machine learning method and the statistic method are greatly affected by the quality of the samples. Due to the large area of the study, it is impossible to record all floods, and thus, the sample quality may not be high enough. (2) The flood susceptibility maps obtained using these methods are semi-quantitative and static. They cannot provide the detailed information on floods, such as flow and submergence range, which can be output by hydro-physical models (Dottori et al., 2016;Hoch and Trigg, 2019). (3) The predictions of this study did not consider the impact of future global climate change on floods which is a popular trend in current research. (4) For the index system, due to the limitations of the data availability, it is difficult for this index system to cover all of the factors that affect the occurrence of floods, e.g., flood control projects such as check dam. In future research, a more comprehensive indicator system needs to be established. Moreover, considering climate change, researchers can try to combine machine learning methods and physical models to obtain more accurate dynamic results in the future.

### 5 Conclusions

In this study, based on 11 flood condition factors, we adopted a machine learning method (SVM) to generate a flood susceptibility map for the Belt and Road region. Based on the spatial distribution of the flood susceptibility, the areas with the highest and high flood susceptibility accounted for 12.22% and 9.57% of the total study area, respectively, and these areas are mainly distributed in the southwestern part of Eastern Asia and almost all of Southeast and South Asia. Moreover, the spatial distribution of the flood susceptibility in Eastern Asia and the CEE has a clear regularity, decreasing from southeast to northwest and from south to north, respectively. According to calculated FSCI values, of the 66 countries in the



study area, 16 had the highest flood susceptibility level and 5 countries had a high flood susceptibility

level. Southeast Asia is considered as the most serious region, with the highest FSCI (4.49). Southeast

Asia contains nine countries with the highest flood susceptibility level and two countries with a high

flood susceptibility level. South Asia suffers from serious threat of flooding, second only to Southeast

Asia. Five of the eight countries in South Asia have the highest flood susceptibility level. In addition,

European countries are also facing the possibility of flooding, especially those along the Mediterranean

coast. These regions and countries should pay more attention to the prevention and management of flood

disasters. Thus, the results of this study provide a scientific basis for disaster prevention and mitigation

and for policy planning in the Belt and Road region. Furthermore, the models and index system used in

this study provide a reference for flood susceptibility assessment in large-scale area. Still, some

limitations exist in this paper, and a more comprehensive indicator system, higher quality flood points,

and climate change factors should be considered in future studies.


*Data availability*. Flood inventory map are available at http://floodobservatory.colorado.edu/ (April

2020). DEM data are available at https://srtm.csi.cgiar.org/srtmdata/ (last access: September 2017).

Precipitation          data          (GPM)          are          available          at

https://disc.gsfc.nasa.gov/datasets/GPM_3IMERGDE_06/summary?keywords=GPM/    (last    access:

October 2019). River Density data are available at https://www.openstreetmap.org/ (last access:

December    2019).    Land    cover    data    (MCD12Q1)    are    available    at

https://ladsweb.modaps.eosdis.nasa.gov/search/ (last access: December 2019). Fractional vegetation

cover data (MOD13Q1) are available at https://ladsweb.modaps.eosdis.nasa.gov/search/ (last access:

December 2019). Impervious surface data are available at https://ghslsys.jrc.ec.europa.eu/ (last access:

December 2019). Soil texture data are available at http://www.fao.org/ (last access: December 2019).

*Author contributions*. JL, YL, and YFH were responsible for the collection and processing of the dataset.

JL and JNX conceptualized the study and developed the methodology. JL and JNX were responsible for

the analysis and validation of the results, and finished the original draft preparation. JL, JNX, WMC,

YFC, YD, WH, GY all participated in the reviewing of methodology, results, and article. All authors

contributed to paper preparation and agreed to the published version of the manuscript.



*Competing interests.* The authors declare that they have no known competing financial interests or personal relationships that could have appeared to influence the work reported in this paper.


*Acknowledgments.* This paper is supported by Strategic Priority Research Program of the Chinese Academy of Sciences (Grant No. XDA20030302), Key R & D project of Sichuan Science and Technology Department (Grant No. 21QYCX0016), National Flash Flood Investigation and Evaluation Project (Grant No. SHZH-IWHR-57), National Key R&D Program of China (2020YFD1100701), and

the Science and Technology Project of Xizang Autonomous Region (Grant No. XZ201901-GA-07), Project form Science and Technology Bureau of Altay Region in Yili Kazak Autonomous Prefecture. The authors are grateful to this support.

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
