# Peer review of "Assessment of Flood Susceptibility Using Support Vector Machine in the Belt and Road Region"

_Natural Hazards and Earth System Sciences, 2021_

## Author Comment (AC1)

We would like to thank Referee #1 for his positive, thoughtful and constructive comments. Here, we will proceed to the responses to questions and comments and outlined the changes made point by point below.

*1. Flood observations of this provenance over 20 years are not good enough to be skillful in the characterisation of flood hazard at this scale. Just because a flood hasn't been recorded since the year 2000, doesn't mean a flood did not happen (within the time period, or indeed before or after it). All your model has therefore done is replicate where DFO/EM-DAT has recorded floods in the past 20 years, having been trained on where DFO/EM-DAT has recorded floods in the past 20 years.*

**Response:**

In flood susceptibility mapping, collecting historical flood events is a crucial step, given the fact that the mapping of flood susceptibility is based on the statistical assumption that future flood will happen in areas having the same conditions which produced them in the past (Costache and Dieu Tien, 2019). Although there is no flood inventory map that can record all floods that have occurred in all years to date, a certain number and accuracy of flood points is sufficient for predicting flood susceptibility. As examples, Bahram Choubin et al. obtained the 51 flood location points identified between the years 2010 to 2017 for an ensemble prediction of flood susceptibility (Choubin et al., 2019). Seyed Vahid Razavi Termeh et al. used the 51 flood locations which were recorded during 2001-2012 for mapping the flood susceptibility in Jahrom Basin (Termeh et al., 2018). The DFO dataset, used in this study, was provided from news, governmental, instrumental, and remote sensing sources, and shows accurate geographical locations flood disasters. This dataset is supported by NASA, the Japanese Space Agency, and the European Space Agency, and is widely used worldwide (Li et al., 2019a). Therefore, the data quality of the DFO dataset is considered to be acceptable. In the current study, a total of 1500 flood points during 2000-2020 were obtained from the DFO dataset. In order to align the time scale of the occurrence of the used flood points with the time scale of precipitation data as much as possible, we therefore only obtained flood points from the year 2000 onwards.

*2. The skill score (AUC) is extremely high, but it is skillfully replicating something that is neither useful nor interesting: GIGO.*

**Response:**

The AUC has been widely used to evaluate the accuracy of the models in machine learning based flood susceptibility mapping studies (Li et al., 2019b; Costache and Dieu Tien, 2019; Termeh et al., 2018; Choubin et al., 2019). The high values of the AUC in this study can show that the good performance of the model and the reliability of results. Based on this, we can do the next step of analysis, which is the focus of this paper.

***3.** It is therefore not clear how any of the data produced in this study advance our understanding of flood hazard, or present novel methods capable of doing so.*

**Response:**

Based on the 1500 flood points and 11 flood conditioning factors, the flood susceptibility map of the Belt and Road region was obtained by a machine learning model (SVM) in this study. This map provides a clear spatial distribution of flood susceptibility in the Belt and Road region. In addition, based on this map, we introduced the FSCI to quantify the degree of flood susceptibility in 7 sub-regions and 66 countries and regions along the Belt and Road, which has not been quantified in this way in previous flood susceptibility studies. For the FSCI, it can be used to quantify the hazard susceptibility of an area of a certain size, such as an administrative unit or a watershed. Thus, the FSCI has the potential to be applied in future studies.

Indeed, this study is not a study that explores the mechanism of flood occurrence. Our aim is to quantify and analyze the spatial distribution pattern of flood susceptibility and the degree of flood risk of countries along the Belt and Road through reliable models and historical disaster data. Therefore, we further analyzed the spatial distribution characteristics and patterns of flood susceptibility in the seven sub-regions, which provides a reliable scientific reference for flood prevention and mitigation in the Belt and Road region. The above results are described in detail in sections 4.2 and 4.3 of the manuscript.

***4.** The writing is generally of poor quality – for which I am sympathetic to the authors – but it does make the manuscript difficult and/or unpleasant to read in some places.*

**Response:**

I am sorry that this article did not give you a good reading experience. As per your comments, we hired a professional language editor to improve the manuscript of its grammatical, typological, and structural problems in the revised article.

***5.** The paper is too long, containing much superflueous information, much of which is*

*incorrect anyway (for instance, claiming the GDP ($2.3bn) of a region spanning 3 continents being roughly equivalent to that of a small English market town).*

**Response:**

Although it is regrettable that there is such an error in the manuscript, it does come from the description of Zhang et al. (Zhang, 2018). We made the following changes to the description in the manuscript.

- It contains a population of about 4.4 billion people accounting for 63% of the world totals.

*6. It is overloaded with references that are not needed, which are often unrelated to the point being made.*

**Response:**

Thank you for your reminder. We have checked the use of citations again in the revised article.

*7. I personally have never heard of the "Belt and Road region", and so would refer to the study area as something else (or make it less arbitrary). Indeed, there's no reason not to deploy this method globally, given its simplicity.*

**Response:**

To strengthen the ties between Asia, Europe, and Africa three continents, "The Belt and Road" Initiative was proposed by China in 2015. The Belt and Road region has been used as a specific study area in many studies (Zhou et al., 2021; Zhou et al., 2020; Shao et al., 2018; Jing et al., 2020; Dong et al., 2018). As the longest economic corridor around the world to date, the Belt and Road region is deeply threatened by floods. According the statistic, during 2000-2020, nearly half (44.9%) of global floods occurred in this region. Therefore, it is of great significance to determine the flood-prone areas in the Belt and Road region in advance and quantify the flood susceptibility level of each country and region for the flood prevention and mitigation.

**References**

Choubin, B., Moradi, E., Golshan, M., Adamowski, J., Sajedi-Hosseini, F., and Mosavi, A.: An ensemble prediction of flood susceptibility using multivariate discriminant analysis, classification and regression trees, and support vector machines, Science of the Total Environment, 651, 2087-2096, https://doi.org/10.1016/j.scitotenv.2018.10.064, 2019.

Costache, R. and Dieu Tien, B.: Spatial prediction of flood potential using new ensembles of bivariate statistics and artificial intelligence: A case study at the Putna river catchment of Romania, Science of the Total Environment, 691, 1098-1118, https://doi.org/10.1016/j.scitotenv.2019.07.197, 2019.

Dong, T. Y., Dong, W. J., Guo, Y., Chou, J. M., Yang, S. L., Tian, D., and Yan, D. D.: Future temperature changes over the critical Belt and Road region based on CMIP5 models, Adv. Clim. Chang. Res., 9, 57-65, https://doi.org/10.1016/j.accre.2018.01.003, 2018.

Jing, C., Tao, H., Jiang, T., Wang, Y. J., Zhai, J. Q., Cao, L. G., and Su, B. D.: Population, urbanization and economic scenarios over the Belt and Road region under the Shared Socioeconomic Pathways, Journal of Geographical Sciences, 30, 68-84, https://doi.org/10.1007/s11442-020-1715-x, 2020.

Li, X., Yan, D., Wang, K., Weng, B., Qin, T., and Liu, S.: Flood Risk Assessment of Global Watersheds Based on Multiple Machine Learning Models, Water, 11, 18, https://doi.org/10.3390/w11081654 2019a.

Li, X. N., Yan, D. H., Wang, K., Weng, B., Qin, T. L., and Liu, S. Y.: Flood Risk Assessment of Global Watersheds Based on Multiple Machine Learning Models, Water, 11, 18, https://doi.org/10.3390/w11081654, 2019b.

Shao, Z.-Z., Ma, Z.-J., Sheu, J.-B., and Gao, H. O.: Evaluation of large-scale transnational high-speed railway construction priority in the belt and road region, Transportation Research Part E-Logistics and Transportation Review, 117, 40-57, https://doi.org/10.1016/j.tre.2017.07.007, 2018.

Termeh, S. V. R., Kornejady, A., Pourghasemi, H. R., and Keesstra, S.: Flood susceptibility mapping using novel ensembles of adaptive neuro fuzzy inference system and metaheuristic algorithms, Science of the Total Environment, 615, 438-451, https://doi.org/10.1016/j.scitotenv.2017.09.262, 2018.

Zhang, J.: The Current Situation of the One Belt and One Road Initiative and Its Development Trend, Annual Report on the Development of the Indian Ocean Region (2017), https://doi.org/10.1007/978-981-13-2080-4_4, 2018.

Zhou, J., Jiang, T., Wang, Y., Su, B., Tao, H., Qin, J., and Zhai, J.: Spatiotemporal variations of aridity index over the Belt and Road region under the 1.5 degrees C and 2.0 degrees C warming scenarios, Journal of Geographical Sciences, 30, 37-52, https://doi.org/10.1007/s11442-020-1713-z, 2020.

Zhou, Y., Wang, T., Peng, R. C., and Hu, H. M.: Spatial-Temporal Characteristics and Factors of Agricultural Carbon Emissions in the Belt and Road Region of China, Polish Journal of Environmental Studies, 30, 2445-2457, https://doi.org/10.15244/pjoes/127414, 2021.

---

## Author Comment (AC2)

We would like to thank Referee #2 for his positive, thoughtful and constructive comments. Here, we will proceed to the responses to questions and comments and outlined the changes made point by point below.

**1.** *First, is it reasonable to use a single model to train the data and predict the flood susceptibility, as the study area is spatially vast and including very different geographic regions from the Tibet Plateau to the European Plain and from the Siberia to the deserts of central Asia?*

**Response:**

First, many studies have successfully applied single models in flood susceptibility mapping and achieved excellent performance. For examples, Tehrany et al. used the single support vector machine (SVM) model to predict the flood susceptibility in Kuala Terengganu basin, Malaysia, and the AUC value of success rate and prediction rate of the SVM reached 88.89% and 84.97% respectively (Tehrany et al., 2015). For the vast study area, Li et al. respectively applied 4 single machine learning models to predict the flood risk of global watersheds, and showed that the 4 models all have good predictive performance (Li et al., 2019). Zhao et al. have successfully used the random forest (RF) model to map flood susceptibility in mountainous areas on a national scale in China, and the RF also obtained good performance with the AUC value of 0.838 (Zhao et al., 2018). Liu et al. have assessed the storm flood risk of Asia using the analytic hierarchy process (AHP), which also showed good model accuracy (Liu et al., 2017). The above studies have shown that a single model has reliable performance in predicting flood susceptibility in a vast study area.

Second, although the study region is indeed very vast, 11 flood conditioning factors have been used in this study, which describes as comprehensively as possible the disaster-pregnant environment in different parts of the study area from the Tibet Plateau to the European Plain and from the Siberia to the deserts of central Asia. In addition, ROC curve, a popular method of machine learning performance evaluation, have been applied in this study to assess the reliability of the SVM model. The ROC curve obtained in this study indicated that the SVM model has excellent predicting performance with AUC value of success rate and prediction rate reached 0.917 and 0.934 respectively. This result showed that the flood susceptibility map generated in this study is reasonable.

Third, although only the flood susceptibility map obtained by the SVM model alone was depicted in this manuscript, several other machine methods were employed

during the experiments, including logistic regression (LR) and random forest (RF). The flood susceptibility maps obtained by LR model and RF model were showed in Fig 2 and Fig 3 respectively, and that obtained by SVM model was showed in Fig 1. By applying the Pearson correlation coefficient method, we calculated the correlation coefficients for the maps obtained by the three models, as shown in Table 1. From the table 1, we can see that the three flood susceptibility maps are well correlated, with all correlation coefficients greater than 0.8. This result suggests that the flood susceptibility map obtained by the SVM model are not coincidental. However, according to the results of the AUC values of these 3 models (Table 2), SVM has the largest AUC value of prediction rate (0.934), followed by RF (0.930) and LR (0.916). Therefore, the map of the SVM model with the highest prediction accuracy were finally selected as the results of the experiments for further analysis in this study.

[Figure]

Figure 1: Flood susceptibility map obtained by support vector machine model.

[Figure]

Figure 2: Flood susceptibility map obtained by logistic regression model.

[Figure]

Figure 3: Flood susceptibility map obtained by random forest model.

Table 1: Pearson correlation coefficients of the three flood susceptibility maps obtained by SVM, LR and RF model.

| Factors | SVM | LR | RF |
|---------|-----|-----|-----|
| SVM | 1 | 0.964 | 0.804 |
| LR | 0.964 | 1 | 0.844 |
| RF | 0.804 | 0.844 | 1 |

Table 2: The AUC values of SVM, LR and RF models.

| Factors | SVM | LR | RF |
|---|---|---|---|
| AUC | 0.934 | 0.916 | 0.930 |

*2. Second, is the data quality of the DFO dataset acceptable? I see the authors mentioned that in the 4.4 section; however, the data quality was not thoroughly evaluated and discussed.*

**Response:**

The DFO dataset was provided from news, governmental, instrumental, and remote sensing sources, and shows accurate geographical locations flood disasters. This dataset is supported by NASA, the Japanese Space Agency, and the European Space Agency, and is widely used worldwide (Li et al., 2019). Therefore, the data quality of the DFO dataset is considered to be acceptable. Despite the good accuracy of this dataset, it still cannot record all the flood events that have occurred in the study area, so in section 4.4 of the manuscript, we consider this as one of the limitations of this study. However, for the time being, this dataset is one of the best datasets for flood susceptibility studies in such vast region.

*3. Third, how were the non-flooded areas/points selected from the DFO dataset? And what do the flooded points and non-flooded points represent? This question determine how we should understand the flood susceptibility.*

**Response:**

For machine learning models, the selection of positive and negative samples is necessary, which is also a requirement for binary analysis (Tehrany et al., 2015). As in previous studies (Costache et al., 2020; Costache and Bui, 2019; Costache and Dieu Tien, 2020; Shafizadeh-Moghadam et al., 2018), in this study, values of 1 and 0 were similarly assigned to the flooded and non-flooded points respectively, which represented the positive and negative samples for the SVM model respectively.

The DFO dataset records the extent of impact of each flood event in the form of vector surface data. After screening out these flood-affected areas, the remaining area in the study area was considered to be free of flooding (Fig. 4). We randomly selected non-flood points in these remaining areas as the negative samples for SVM model. In terms of historical flood points distribution characteristics, the results obtained based

on this non-flood point selection method are somewhat improved compared to the results of Li et al. (Li et al., 2019). Li et al. selected non-flooded points in the deserts and ice fields, which may make the conditions for non-flooding more severe mentioned by themselves.

[Figure]

Figure 4: The distribution of flood and non-flood sample points in the study area.

*4. Fourth, what are the key findings that are novel and instructive from the paper? The Abstract and Conclusion are very general.*

**Response:**

Based on a novel method for non-flood points selection, this study uses a simple and reliable machine learning model (SVM) to assess the flood susceptibility of the Belt and Road region. Then, according to the flood susceptibility map, we introduced the flood susceptibility comprehensive index (FSCI) to quantify the degree of flood susceptibility in 7 sub-regions and 66 countries along the Belt and Road region. The FSCI was calculated as follows, which was also showed in equation (10) in the manuscript.

$$FSCI = \sum_{i=1}^{n} p_i \times \frac{A_i}{S} \tag{1}$$

where *FSCI* is the flood susceptibility comprehensive index of a country; $P_i$ is the class value of the $i_{th}$ flood susceptibility calss; $A_i$ is the areas of $i_{th}$ flood susceptibility class; and *S* is the total area of the country.

In terms of methodology, the non-flood point selection method used in this study

provides useful references for future researchers using the DFO dataset. More importantly, the FSCI proposed in this study to quantify flood susceptibility has the potential to be applied in future studies. The FSCI can be used to quantify the hazard susceptibility of an area of a certain size, such as an administrative unit or a watershed. In terms of the results, the three main results were obtained in this study as follows. (1) The spatial distribution characteristics of flood susceptibility in the Belt and Road region, which was described in section 4.2 of the manuscript. (2) The spatial distribution characteristics of the seven sub-regions of the Belt and Road and the quantification of flood susceptibility, which was analyzed in section 4.3 of the manuscript. (3) The quantification of flood susceptibility in the 66 countries of the Belt and Road region, which was also analyzed in section 4.3 of the manuscript.

Based on the above three results, the main findings of this study are as follows. (1) the areas with the highest and high flood susceptibility accounted for 12.22% and 9.57% of the total study area, respectively, and these areas are mainly distributed in the southwestern part of Eastern Asia and almost all of Southeast and South Asia. (2) Of the seven sub-regions in the Belt and Road region, Southeast Asia is most susceptible to flooding and has the highest FSCI (4.49), followed by Southern Asia and the CEE. (3) According to calculated FSCI values, of the 66 countries in the study area, 16 had the highest flood susceptibility level and 5 countries had a high flood susceptibility level. (3) Of the seven sub-regions in the Belt and Road region, Southeast Asia is most susceptible to flooding and has the highest FSCI (4.49), followed by Southern Asia and the CEE. The above findings not only pinpoint the flood-prone areas, but also identify the countries and regions most affected by flooding with quantitative values (FSCI), which is an improvement compared to previous studies.

Based on the above description, we have changed the abstract and conclusion in the manuscript as follows.

**Abstract:** Floods have occurred frequently all over the world. During 2000-2020, nearly half (44.9%) of global floods occurred in the Belt and Road region because of its complex geology, topography, and climate. However, spatial distribution characteristics of flood susceptibility in the Belt and Road region remains unclear. Here, a database was built in this study containing 11 flood condition factors and 1500 flooded points. Next, we used a novel method to select the same number of non-flooded points for the negative samples. Subsequently, support vector machine (SVM) model was applied to train the samples and predict the flood susceptibility. Finally, the concept of

ecological vulnerability synthesis index in the ecological field was introduced into this study, and the flood susceptibility comprehensive index (FSCI) was proposed to quantify the degree of flood susceptibility of each country and region. The results reveal the following. (1) The SVM model used in this study has an excellent accuracy, and the AUC values of the success-rate curve and prediction-rate curve were higher than 0.9 (0.917 and 0.934 respectively). (2) The areas with the highest and high flood susceptibility account for 12.22% and 9.57% of the total study area respectively, and these areas are mainly located in the southeastern part of Eastern Asia, almost the entirely of Southeast Asia and South Asia. (3) Of the seven sub-regions in the Belt and Road region, Southeast Asia is most susceptible to flooding and has the highest FSCI (4.49), followed by South Asia. (4) Of the 66 countries in this region, 16 of the countries have the highest flood susceptibility level (normalized FSCI > 0.8) and 5 countries (normalized FSCI > 0.6) have a high flood susceptibility level. This study provides scientific references for flood prevention and mitigation in the Belt and Road region, and lays a theoretical basis for the quantification of flood susceptibility.

**Conclusion:**

In this study, we prepared a geospatial database in a first step, which contained 11 flood conditioning factors and 1500 flood locations. Next, based on a novel method of selecting non-flooded points, 1500 non-flooded locations were identified as the negative samples. Then we adopted a machine learning model (SVM) to train the samples and generated a flood susceptibility map for the Belt and Road region. More interestingly, we introduced the FSCI from the concept of ecological vulnerability synthesis index to quantify the flood susceptibility level of 7 sub-regions and 66 countries in the study area. According to the spatial distribution of the flood susceptibility, the areas with the highest and high flood susceptibility accounted for 12.22% and 9.57% of the total study area, respectively, and these areas are mainly distributed in the southwestern part of Eastern Asia and almost all of Southeast and South Asia. Moreover, the spatial distribution of the flood susceptibility in Eastern Asia and the CEE has a clear regularity, decreasing from southeast to northwest and from south to north, respectively. According to calculated FSCI values, of the 66 countries in the study area, 16 had the highest flood susceptibility level and 5 countries had a high flood susceptibility level. Southeast Asia is considered as the most serious region, with the highest FSCI (4.49). Southeast Asia contains nine countries with the highest flood

susceptibility level and two countries with a high flood susceptibility level. South Asia suffers from serious threat of flooding, second only to Southeast Asia. Five of the eight countries in South Asia have the highest flood susceptibility level. In addition, European countries are also facing the possibility of flooding, especially those along the Mediterranean coast. These regions and countries should pay more attention to the prevention and management of flood disasters. Thus, the results of this study provide a scientific basis for disaster prevention and mitigation and for policy planning in the Belt and Road region. Furthermore, the FSCI proposed in this study can be used to quantify the flood susceptibility of an area of a certain size, such as an administrative unit or a watershed. Still, some limitations exist in this paper, and a more comprehensive indicator system, higher quality flood points, and climate change factors should be considered in future studies.

**References**

Costache, R. and Bui, D. T.: Spatial prediction of flood potential using new ensembles of bivariate statistics and artificial intelligence: A case study at the Putna river catchment of Romania, Science of the Total Environment, 691, 1098-1118, http://doi.org/10.1016/j.scitotenv.2019.07.197, 2019.

Costache, R. and Dieu Tien, B.: Identification of areas prone to flash-flood phenomena using multiple-criteria decision-making, bivariate statistics, machine learning and their ensembles, Science of the Total Environment, 712, http://doi.org/10.1016/j.scitotenv.2019.136492, 2020.

Costache, R., Hong, H., and Quoc Bao, P.: Comparative assessment of the flash-flood potential within small mountain catchments using bivariate statistics and their novel hybrid integration with machine learning models, Science of the Total Environment, 711, http://doi.org/10.1016/j.scitotenv.2019.134514, 2020.

Li, X., Yan, D., Wang, K., Weng, B., Qin, T., and Liu, S.: Flood Risk Assessment of Global Watersheds Based on Multiple Machine Learning Models, Water, 11, 18, https://doi.org/10.3390/w11081654 2019.

Liu, J., Wang, X., Zhang, B., Li, J., Zhang, J., and Liu, X.: Storm flood risk zoning in the typical regions of Asia using GIS technology, Natural Hazards, 87, 1691-1707, https://doi.org/10.1007/s11069-017-2843-1, 2017.

Shafizadeh-Moghadam, H., Valavi, R., Shahabi, H., Chapi, K., and Shirzadi, A.: Novel forecasting approaches using combination of machine learning and statistical models for flood susceptibility mapping, Journal of Environmental Management, 217,

1-11, http://doi.org/10.1016/j.jenvman.2018.03.089, 2018.

Tehrany, M. S., Pradhan, B., Mansor, S., and Ahmad, N.: Flood susceptibility assessment using GIS-based support vector machine model with different kernel types, Catena, 125, 91-101, https://doi.org/10.1016/j.catena.2014.10.017, 2015.

Zhao, G., Pang, B., Xu, Z., Yue, J., and Tu, T.: Mapping flood susceptibility in mountainous areas on a national scale in China, Science of the Total Environment, 615, 1133-1142, https://doi.org/10.1016/j.scitotenv.2017.10.037, 2018.